# The Determination of the Inward Leakage through the Skin–Facepiece Interface of the Protective Half-Mask

Tomáš Brestovič [1], Marián Lázár [1,*], Natália Jasminská [1], Jozef Živčák [2], Radovan Hudák [2], Lukáš Tóth [1] and Romana Dobáková [1]

1 Department of Power Engineering, Faculty of Mechanical Engineering, Technical University of Košice, 04200 Košice, Slovakia; tomas.brestovic@tuke.sk (T.B.); natalia.jasminska@tuke.sk (N.J.); lukas.toth@tuke.sk (L.T.); romana.dobakova@tuke.sk (R.D.)
2 Department of Biomedical Engineering and Measurement, Faculty of Mechanical Engineering, Technical University of Košice, 04200 Košice, Slovakia; jozef.zivcak@tuke.sk (J.Ž.); radovan.hudak@tuke.sk (R.H.)
* Correspondence: marian.lazar@tuke.sk; Tel.: +421-55-602-4358

**Abstract:** The present article describes the measurements of flow rates of the inward air leakage through the skin–facepiece interface of a protective half-mask with replaceable filters. The measurements were carried out while applying an indirect method in which the pressure drops in a compressed air container were measured, and subsequently, the total flow rate of the leak was calculated. This methodology facilitated measuring extremely low air flow rates at the atmospheric pressure of $3.2 \times 10^{-6}$ $m^3 \cdot s^{-1}$. A numerical analysis of the inward air leakage through the gaps between the face and the facepiece of the mask was carried out with the aim of identifying the cross-sectional area of the leak. With the tested mask, which was made of Santoprene 8281-45MED, the leakage measured during inhalation was 0.21%, which corresponded to the cross-sectional area of only 0.14 $mm^2$.

**Keywords:** pandemic; COVID-19; personal protective equipment; measurement

## 1. Introduction

The propagation of various infectious diseases at a local level is not an exceptional phenomenon, even in the 21st century. Gastrointestinal infections and acute inflammatory diseases of the respiratory system that are caused, for example, by influenza, rhinoviruses, coronaviruses, etc., occur on a regular basis or sporadically in both developed as well as less developed communities. Therefore, life-threatening infections still represent a danger in various circumstances. Such a danger is even strengthened by the natural development and mutations of various bacteria, viruses, and fungi, as well as by climate change and globalization [1–3].

Globally, the way out from the fight with infections is the development of efficient medications and vaccines, which is a very demanding process with regard to the duration and cost. While these medications mostly mitigate the course and consequences of diseases, efficient vaccines provide a long-lasting protective effect if they are applied to a sufficiently large population. In many cases, vaccines result in the elimination of a particular infection. Prevention represents an essential part of the efforts aimed at suppressing pandemics. Strict adherence to fundamental hygiene habits and the use of the relevant personal protective equipment (PPE) are also efficient means to reduce the risk of infection. Such PPE includes face masks, respirators, masks, protective goggles, face shields, gloves, etc. [4,5].

The current global pandemic of the SARS-CoV-2 virus revealed insufficient readiness in health protection and prevention of the COVID-19 respiratory disease, which was first detected in the Chinese town of Wuhan. The size of the virus causing COVID-19 has been determined under the Transmission Electron Microscope (TEM) as 60–140 nm, which

averages to 100 nm [6]. This size is similar to the size of the SARS coronavirus, which is also 100 nm [7]. The virus gains entry through the mucous membranes, and the main routes of transmission include: virus-infected droplets and aerosols, or a contact with skin, surfaces, or other items contaminated by fomites [8–11]. As of the end of July 2021, COVID-19 globally infected as many as 199,194,369 people, out of whom 4,241,981 died.

Research, development, and testing of novel materials to be used in air filtration and the manufacture of respirators with replaceable filters, which facilitate efficient protection against the SARS-CoV-2 virus, are still regarded as highly topical due to persisting problems with the present pandemic. The 3D technology was used in the production of PPE by the team of scientists, Thierry et al. [12]. The adaptors and the clip were printed using the fused deposition modelling. The results obtained by this research team were transformed into the EasyBreath mask, which was tested in an operating theatre. When compared to the combined use of a headshield and an FFP2/N95 facemask, surgeons reported they felt safer and had better vision of the surgical field thanks to the portable light. The case study demonstrated a new approach to designing and manufacturing PPE in emergency situations that relies on the rapid development and domestic manufacturing of products through 3D printing technologies. The same topic was also discussed in the paper by Sterman et al. [13]. A team led by Leung focused their attention on the development of a novel charged PVDF nanofibre filter technology in order to effectively capture the fast-spreading, deadly airborne coronaviruses, especially COVID-19, with the target aerosol size of 100 nm [7]. The virus and its attached aerosol were simulated by sodium chloride aerosols sized 50–500 nm, which were generated from a sub-micron aerosol generator. Filter fibres were subsequently electrostatically charged by a corona discharge. This team designed four new filters in total, whereas one of the filters achieved 90% efficiency and exhibited an ultralow pressure drop of only 18 Pa (1.9 mm water), and another filter met the 30 Pa threshold and exhibited a high efficiency of 94%. Generally, newly developed products may only be marketed if they are certified and tested in accredited laboratories.

The highest degree of protection of the respiratory system against infections is provided by products classified as FFP3 protective equipment. Respirators, half masks, and masks that belong to this category must meet a vast number of requirements prescribed by the EN 143, EN 149, and EN 1827 standards, as well as by the relevant standards applicable to respirators protecting against particles. The basic tests which are mandatory for the PPE include the tests for the particle penetration through filters, the concentration of carbon monoxide in the inhaled air, respiratory resistance, filter clogging with dolomite dust, total penetration through the mask, combustibility, skin tolerance of the mask, etc.

Due to only a limited number of accredited laboratories, long waiting periods, and a lack of certified testing equipment available on the market, there is a need for the development of proprietary measuring apparatuses that would facilitate simple and less costly testing of respirators, half masks, and their accessories prior to the certification process. The present article discusses in more detail a new design of a measuring apparatus intended for identification of the air penetration through the skin–facepiece interface of a half mask, and the results of the testing carried out with this newly designed measuring apparatus.

## 2. A Design of the Experimental Equipment

The line of the facepiece of the half mask must copy, as accurately as possible, the wearer's face. Due to variability of human faces, the mask must exhibit hardness and flexibility that will ensure that it optimally fits. Despite the efforts aimed at designing an optimal shape of the facepiece of the half mask, during breathing, the air was still leaking inside the mask through the gaps which may be formed between the mask and a human face. The measurements of the total inward leakage were carried out in a chamber while the mask was actually applied on a control object, and a focus of the analysis was the inward leakage of sodium hydroxide aerosol.

The certification of half masks requires the determination of the total inward leakage of aerosol through all connections and contact surfaces of a mask. According to the EN 1827 + A1 standard, the inward leakage of the test aerosol through the valves (if on the mask) and the skin–facepiece interface of the mask must not exceed 2%. In order to create an analogue measurement, but without the necessity for using aerosol, an experimental stand was constructed and used for the measurements of the total air flow through the leaks in the facepiece area of the mask, and it was proportionally compared to the total air flow during inhalation. The key hypothesis of the analogy was that there was equality between the total penetration of aerosol and the ratio of air leakage to the total air flow during inhalation or exhalation. In order to create the most unfavourable conditions in which the air leakage was at a maximum level, it was necessary to create adequate negative pressure during inhalation and positive pressure during exhalation. According to the EN 1827 standard, the maximum negative pressure during inhalation is 420 Pa at the air flow rate of 95 L·min$^{-1}$ for particle-filtering masks of the FMP3 class. During exhalation at the flow rate of 160 L·min$^{-1}$, positive pressure in the mask must not exceed 300 Pa.

The target parameter of the measuring stand was the determination of leakage during inhalation below 0.2%; at the flow rate during inhalation of 95 L·min$^{-1}$, it was therefore necessary to measure the leak flow rate of 0.19 L·min$^{-1}$ (or 3.2 mL·s$^{-1}$). Measuring such a low flow rate at pressures approaching the atmospheric pressure is significantly difficult. That is why a measuring system was designed with an indirect measuring method, i.e., measuring the air pressure in the container of a defined size (Figure 1). With a known volume of the container and actual changes in the pressure over time, it was possible to identify the development of inward leakage through the facepiece over time.

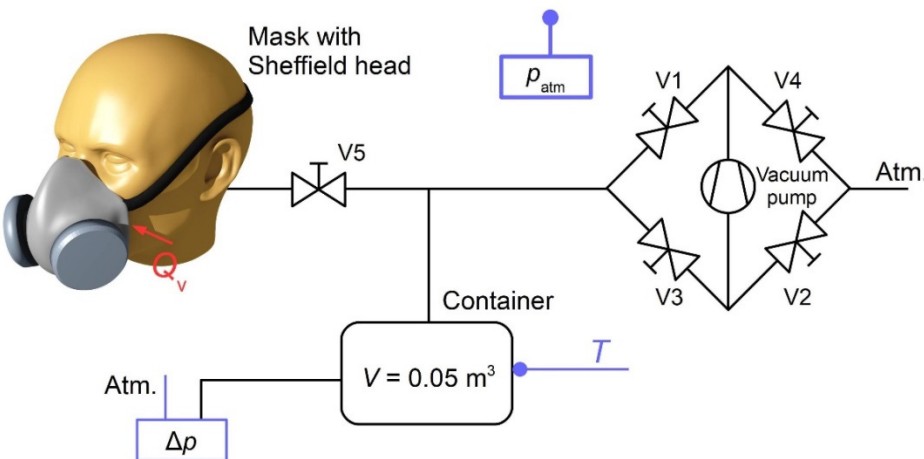

**Figure 1.** Scheme of individual components of the system measuring the respirator inward leakage in the facepiece area.

During the measurements, an analysis was focused on the isothermal pressure drop in the container, which was caused by changes in the weight of the air. The following equation was used:

$$\mathrm{d}m = \frac{V}{r \cdot T}\mathrm{d}p \quad (\mathrm{kg}) \tag{1}$$

wherein d$m$ is the elementary change in the weight of the air in the system, which was caused by the leakage (kg); $V$ is the total internal volume of the measuring system, including the distribution systems and the internal volume of the mask (m$^3$); $r$ is the specific gas constant (J·kg$^{-1}$·K$^{-1}$); $T$ is the average temperature of the measurement system (K); and d$p$ is the elementary change in the air pressure (Pa).

The mass flow rate of the air leaking through the respirator, as divided by time differential d$\tau$, was calculated using the following formula:

$$Q_m = \frac{V}{r \cdot T} \cdot \frac{\mathrm{d}p}{\mathrm{d}\tau} \quad (\mathrm{kg \cdot s^{-1}}) \tag{2}$$

The volumetric flow rate of the air leaking through the respirator was calculated by dividing the mass flow rate by the gas density $p/(r \cdot T)$:

$$Q_V = \frac{V}{p} \cdot \frac{\mathrm{d}p}{\mathrm{d}\tau} \quad (\mathrm{m}^3 \cdot \mathrm{s}^{-1}) \tag{3}$$

After rewriting the derived notation into a differential notation, the resulting formula for the calculation of the leaking air flow rate was as follows:

$$Q_V = \frac{V}{p} \cdot \frac{\Delta p}{\Delta \tau} \cdot 6 \times 10^4 \quad (\mathrm{L} \cdot \mathrm{min}^{-1}) \tag{4}$$

The experimental measuring apparatus which was used for the identification of the air leakage through the skin–facepiece interface of the half mask consisted of the membrane vacuum pump of the N86KN.18 type, an adjusted pressure container, a distribution system with valves (Swagelok Instrument Plug Valve-SS-6P4T-MM-BK), and the Sheffield head (Figure 1). Increases and decreases in pressure in the expansion vessel were scanned using a differential pressure sensor, DPS 300 816-CS2M-1-0-8-0-1-Y00-M-000. The measurements of the air temperature were carried out using a resistance temperature sensor, Pt100. The measured values were processed using a universal 9-input measuring instrument, Almemo 2890-9. The results of the measurements were evaluated using a created computer software, which was created in the C ++ programming language by an employee of the Department of Energy Engineering.

The assembled measuring apparatus was used to test the half mask without an exhalation valve, which was designed and produced at the Faculty of Mechanical Engineering of the Technical University of Košice (Figure 2). The shape of the respirator facepiece was optimised on the basis of the results of similarly conducted analyses of biological parameters of more than 20 scans of human faces. The output of these analyses was the elimination of the risk of potentially infecting wearers of this PPE as a result of particle leakage around the filter.

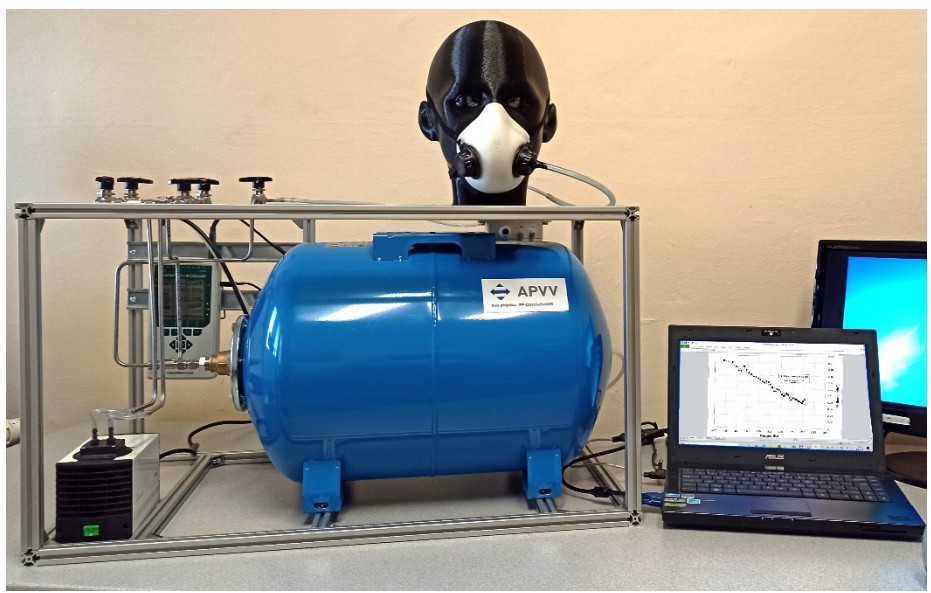

**Figure 2.** A view on the measuring equipment.

As a filter housing material, the high reusability polyamide PA 12 was used, which is a robust thermoplastic that produces high-density parts with balanced property profiles and strong structures. This material is ideal for complex assemblies, housings, and watertight applications, and is also biocompatible (USP Class I-VI and US FDA guidance for Intact Skin Surface Devices). The body of the half mask and the contact surface of the skin–

facepiece interface were made of Santoprene 8281-45MED. The purpose was to maximise the adhesion of the face-touching surface of the half mask to the skin and minimise the skin irritation in the area where the wearer's face contacts the half mask. Santoprene 8281 MED (ExxonMobil) belongs to the soft, colourable, non-hygroscopic thermoplastic vulcanizate (TPV) in the thermoplastic elastomer (TPE) family for medical and healthcare applications.

The determination of the inward leakage through the skin–facepiece interface of the half mask was based on indirect measurements of the flow rate of the atmospheric air. In a pressure container, a membrane vacuum pump of the N86KN.18 type was used to create positive pressure, or negative pressure, of 600 Pa by opening valves V1 and V2, or V3 and V4 (Swagelok Instrument Plug Valve-SS-6P4T-MM-BK). The air flow between the vacuum pump and the pressure container was stopped by closing valves V1 through V4 (Figure 3). Subsequently, after opening valve V5 (Swagelok Instrument Plug Valve-SS-6P4T-MM-BK), which regulated the air flow from the pressure container towards the tested half mask, an indirect measurement of inward leakage through the skin–facepiece interface of the tested half mask was carried out. The tested half mask was placed on the Sheffield head, which was connected to the pressure container via a flexible hose. During the measurement, the openings in the half mask for a pair of bayonet filters were hermetically closed and sealed with s silicone sealant. The air leaking inside the half mask during the measurement through the skin–facepiece interface caused changes in the pressure in the system over time. The calculations made using formula (4) facilitated producing the curve of a correlation between the inward leakage and the differences between the air pressure inside the mask and the air pressure outside the mask.

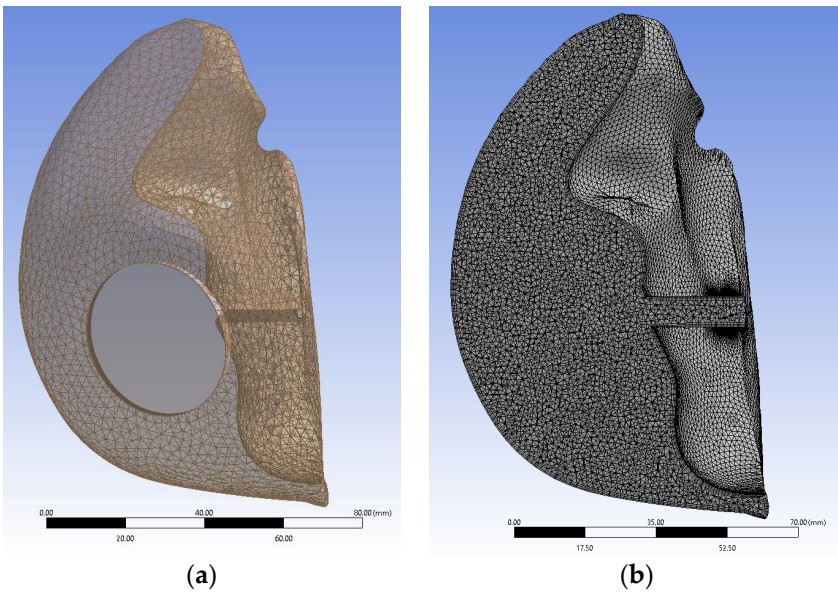

(**a**)    (**b**)

**Figure 3.** (**a**) Illustration of the examined domain and (**b**) a cross-section of the domain in the vertical plane across the created mesh.

## 3. Defining the Total Cross-Sectional Area of the Leaks of the Mask

A size of the total cross-sectional area of the leak significantly affected the total air flow. For the purpose of calculating the air flow, it was first necessary to define velocity of the leaking air with a known value of the pressure difference, based on the Bernoulli's principle, whereas a constant air density was assumed for minor changes in the absolute pressure. The equation was as follows:

$$v = \varphi \cdot \sqrt{\frac{2 \cdot \Delta p}{\rho}} \ (\text{m·s}^{-1}) \tag{5}$$

wherein $v$ is the velocity of the leaking air (m·s$^{-1}$); $\Delta p$ is the pressure difference (Pa); $\rho$ is the air density (kg·m$^{-3}$); and $\varphi$ is the coefficient of velocity. The coefficient of velocity ranged from 0.8 to 0.98, depending on the shape of the gap between the human face and the facepiece of the mask. The total cross-sectional area of the leak was identified using the following formula for the air flow:

$$S = \frac{Q_v}{\mu \cdot \sqrt{\frac{2 \cdot \Delta p}{\rho}}} \quad (\text{m}^2) \tag{6}$$

wherein $S$ is the total cross-sectional area of the leak (m$^2$); $Q_v$ is the volumetric air flow through the leak (m$^3$·s$^{-1}$); and $\mu$ is the efflux coefficient (1).

In order to identify the coefficient of velocity and the efflux coefficient, depending on the total cross-sectional area of the leak, numerical simulations of the air flow during inhalation were carried out. The examined domain was the space between the Sheffield head and the model of the mask with two replaceable filters (Figure 3). Figure 3a shows the real geometry of the mask in stl. format, while Figure 3b shows a cross-section of the domain in the vertical plane across the created mesh.

The mesh of the designed geometry consisted of 1 million elements, and near the wall, the mesh was denser so that the thickness of the first element reached a dimensionless distance of $y^+ = 40$. The flowing medium was the air with a temperature of 25 °C, and its density was calculated using an equation of state. At the site of the Sheffield head mouth, an output boundary condition was defined, i.e., the flow rate of 95 L·min$^{-1}$ during inhalation. In lateral spaces of the mask, there was a pair of filters. At these sites, an input boundary condition was applied with a defined static pressure, while considering the maximum negative pressure of $-420$ Pa during inhalation (applicable to FMP3 according to EN 1827). At the height of the mouth, leaks were created on both sides with the total cross-sectional area that ranged from 0.05 to 5 mm$^2$. The leaks simulated an imperfect contact between the skin and the facepiece of the mask through which the external air was flowing into the internal space of the mask. At these sites, boundary conditions were defined with the total relative pressure of 0 Pa. Figure 4 shows the velocity contours in the horizontal plane of the mask, which crossed the centre of the mouth and the lateral leaks, while the total cross-sectional areas were 0.5 and 5 mm$^2$.

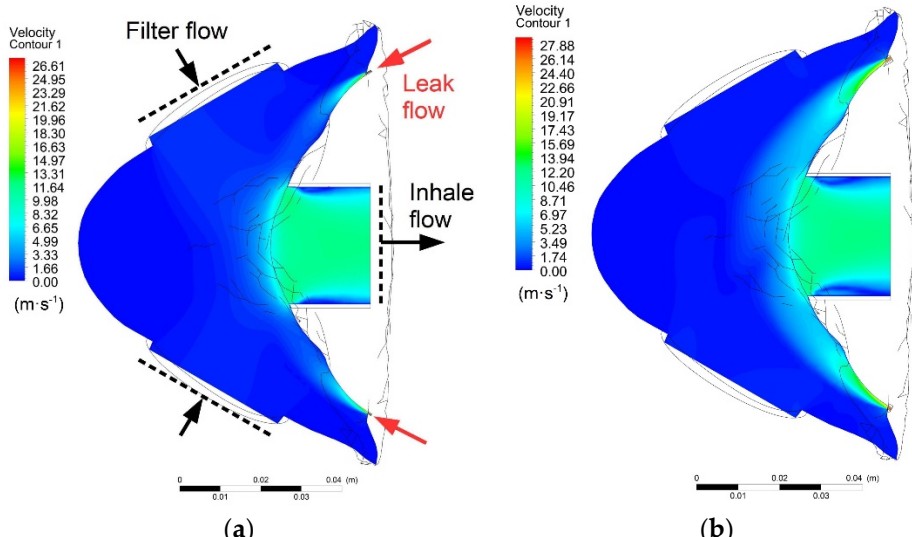

**Figure 4.** (**a**) Velocity contours for the leak with the cross-sectional area of 0.5 mm$^2$ and (**b**) 5 mm$^2$.

Figure 5 shows the flow lines of the inhaled air. The blue lines are the flow lines of the air passing through filters, and the red lines are the flow lines of the air inhaled through the facepiece leaks.

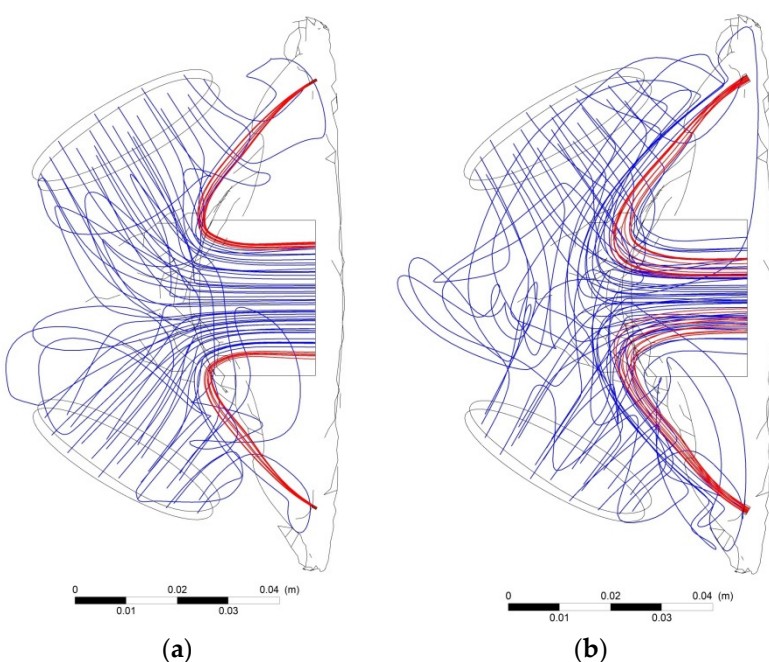

**Figure 5.** (**a**) Flow lines of the inhaled air with the leak areas of 0.5 mm$^2$ and (**b**) 5 mm$^2$.

Static pressure inside the mask was constant in all simulations. It was therefore possible to calculate the theoretical (maximum) velocity of the air flowing through the leaks using formula (5) with the coefficient of velocity equal to 1. The theoretical velocity was 26.62 m·s$^{-1}$, and when this velocity value was multiplied by the cross-sectional area of the leak, the result was the theoretical air flow through the leak $Q_{v\text{-theor}}$. After identifying the true velocity and flow rate using numerical calculations of the air flow, it was possible to calculate the coefficient of velocity and the efflux coefficient. The resulting data are listed in Table 1.

**Table 1.** Calculated parameters corresponding to the total cross-sectional area of the leak of the mask.

| S (mm$^2$) | $v$ (m·s$^{-1}$) | $Q_v$ (m$^3$·s$^{-1}$) | $Q_{v\text{-theor.}}$ (m$^3$·s$^{-1}$) | $\varphi$ (1) | $\mu$ (1) | Leak (%) |
|---|---|---|---|---|---|---|
| 0.05 | 23.62 | $1.023 \times 10^{-6}$ | $1.331 \times 10^{-6}$ | 0.887 | 0.770 | 0.065 |
| 0.1 | 23.82 | $2.382 \times 10^{-6}$ | $2.662 \times 10^{-6}$ | 0.895 | 0.895 | 0.150 |
| 0.5 | 25.17 | $1.258 \times 10^{-5}$ | $1.331 \times 10^{-5}$ | 0.946 | 0.945 | 0.795 |
| 1 | 25.99 | $2.600 \times 10^{-5}$ | $2.662 \times 10^{-5}$ | 0.976 | 0.977 | 1.681 |
| 5 | 26.15 | $1.307 \times 10^{-4}$ | $1.331 \times 10^{-4}$ | 0.982 | 0.982 | 8.406 |

The curves of changes in the coefficient of velocity and efflux coefficient are shown in Figures 6 and 7, where the curves may be approximated using the sigmoid function. The best results of regression were achieved while using the Boltzmann function described by Equations (7) and (8).

$$\varphi = 0.982 - \frac{0.1474}{1 + e^{\frac{S - 0.2033}{0.2621}}} \quad (-) \tag{7}$$

Equation (7) was used to identify the coefficient of velocity for a variable value of the leak area, which may be used to identify the true velocity of the inhaled air.

$$\mu = 0.98 - \frac{3653.7}{1 + e^{\frac{S + 0.9778}{0.1052}}} \quad (-) \tag{8}$$

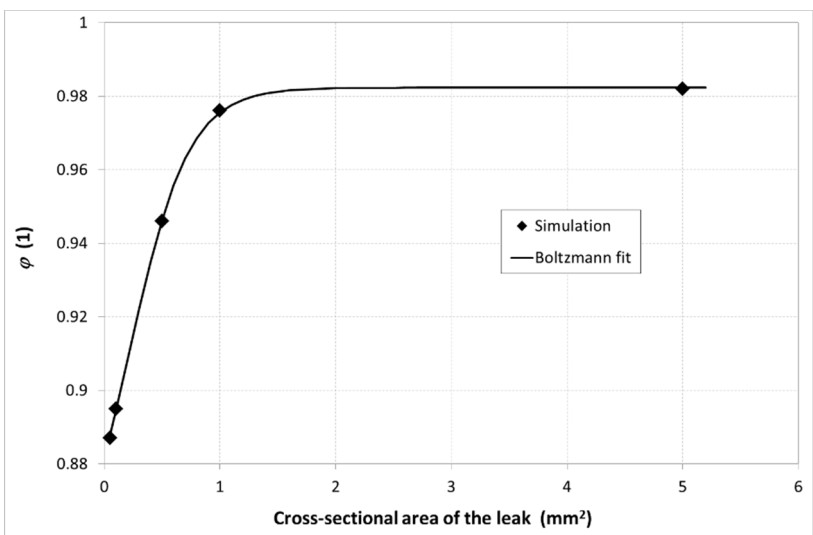

**Figure 6.** Curve of the correlation between the flow rate coefficient and the cross-sectional area of the leak.

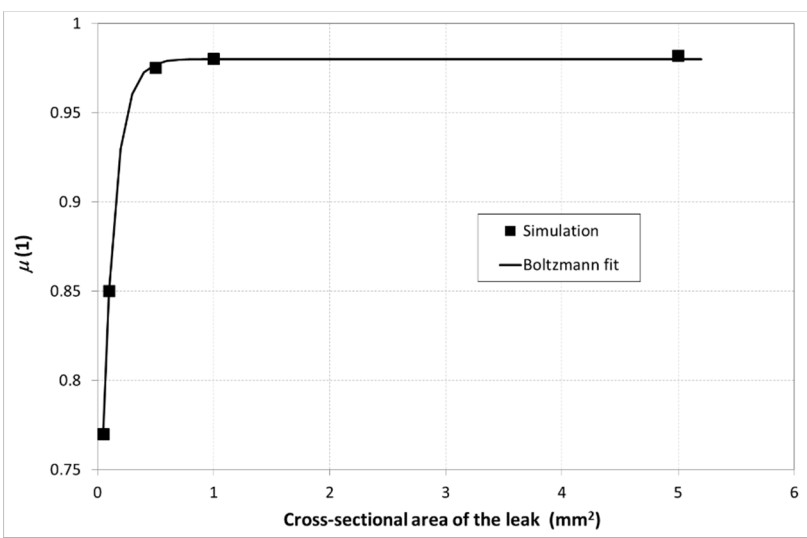

**Figure 7.** Curve of the correlation between the efflux coefficient and the cross-sectional area of the leak.

Equation (8) may be used to identify the efflux coefficient, which must be known in order to identify the true flow rate of the air inhaled through the leak. Figure 8 shows the curve of a correlation between the cross-sectional area of the leak and the leak percentage. An equation of the linear regression of the values identified by simulations represents formula (9).

$$S = 0.593 \cdot \varepsilon + 0.0137 \; \left(mm^2\right) \tag{9}$$

wherein $\varepsilon$ is the leak percentage (%).

The curve indicates the maximum cross-sectional area of the leak at which the leak is still below the defined leak threshold. For FMP3 half masks, the inward leakage must not exceed 2%. According to the results of the linear regression, this corresponds to the total cross-sectional area of the leak of 1.2 mm$^2$.

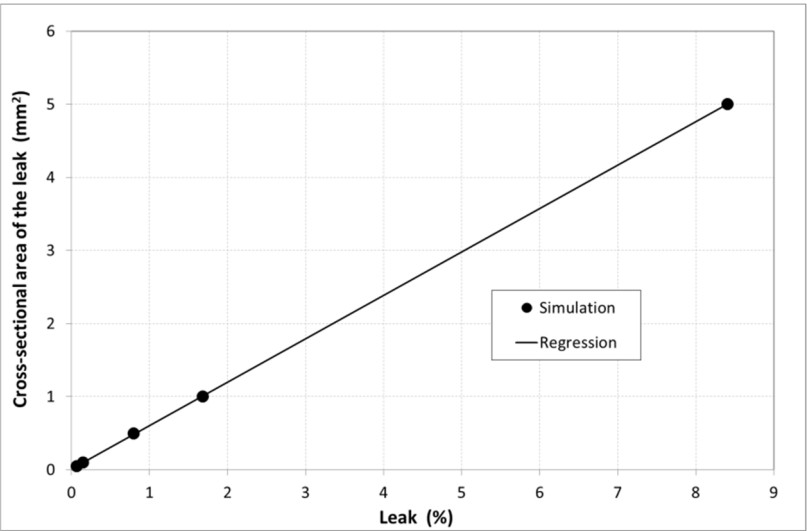

**Figure 8.** Curve of the correlation between the cross-sectional area of the leak and the leak percentage.

### 4. Analysis of the Results and Discussion

Experimental measurements were carried out on a previously produced half mask, which was placed on a Sheffield head. During the first measurement, the flow rates of the air leaking through the skin–facepiece interface during inhalation were analysed. Prior to the measurement, a vacuum pump was used to reduce the pressure in the pressure container to the value of −600 Pa. After the vacuum pump was disconnected by closing valves V1 through V4 and opening valve V5, the pressure in the system gradually increased, as is shown in Figure 9.

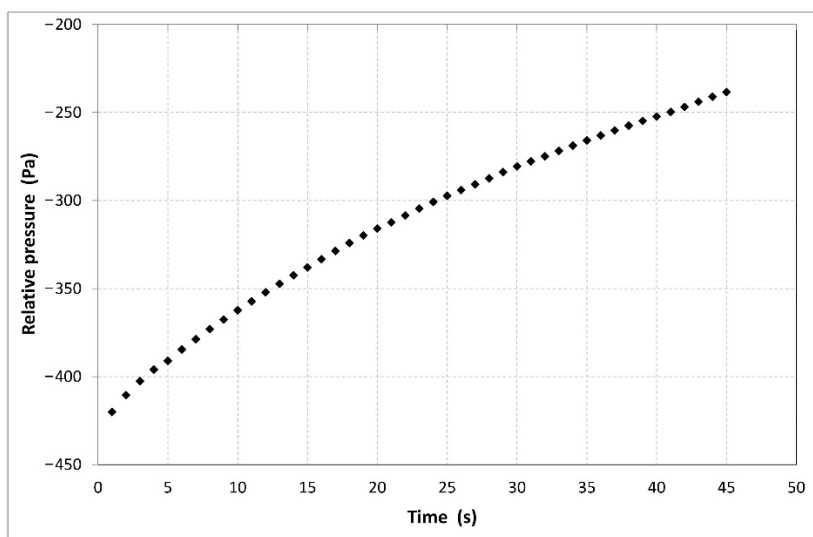

**Figure 9.** Curve of the changes in the pressure in the container over time during the measurement of inward leakage during inhalation.

Due to the pressure drop in the supply pipeline, relative pressure inside the mask was higher than the pressure in the container; however, an analysis revealed that the pressure drop in the distribution pipelines was negligible due to extremely low velocity of the air, and it did not exceed the value of 0.1 Pa. After applying the measured data in formula (4), the curve of the correlation between the leakaging air flow and the negative pressure inside the mask was produced (Figure 10).

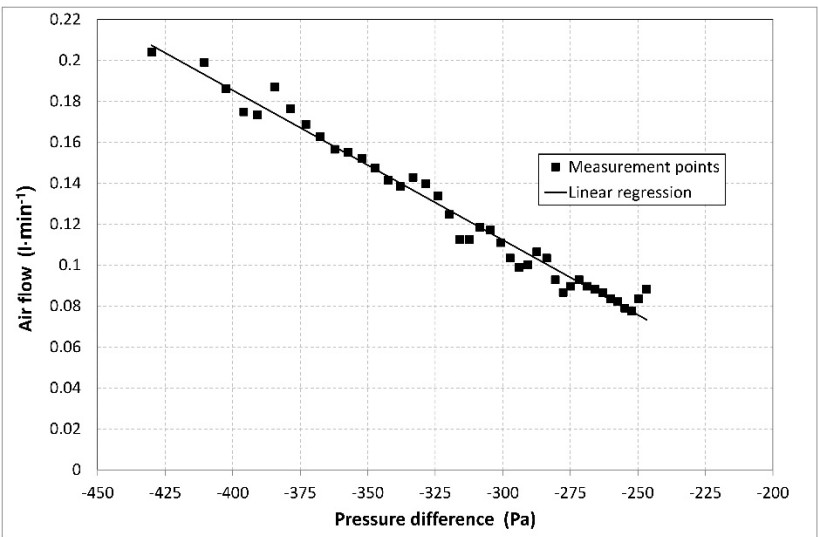

**Figure 10.** A correlation between the leaking air flow rate and negative pressure (inhalation) inside the mask.

At a maximum permissible negative pressure for FMP3 respirators, representing 420 Pa, the flow rate of the air leaking through the mask was 0.2 L·min$^{-1}$. This value represents 0.21% of the flow rate of 95 L·min$^{-1}$ during inhalation, as defined by the EN 1827 standard. According to formula (9), the total cross-sectional area of the leak was only 0.14 mm$^2$.

The total cross-sectional area of all leaks in inhalation may be different from the total cross-sectional area of the leak in exhalation. That is why the second measurement was carried out. The purpose of this measurement was to identify the flow rate of the leaking air during exhalation, at which the air inside the mask exhibits positive pressure compared to the atmosphere. The second measurement differed from the first one in creating a positive pressure of +600 Pa in the pressure container using a vacuum pump, while the air leakage through the skin–facepiece interface during exhalation was measured. The flow rates of the leaking air during exhalation are shown in Figure 11. To facilitate a comparison of the leaks, the curve of the leakage measured during exhalation was supplemented with the data obtained in the first measurement, which was carried out during inhalation, but the pressure was converted into positive absolute values.

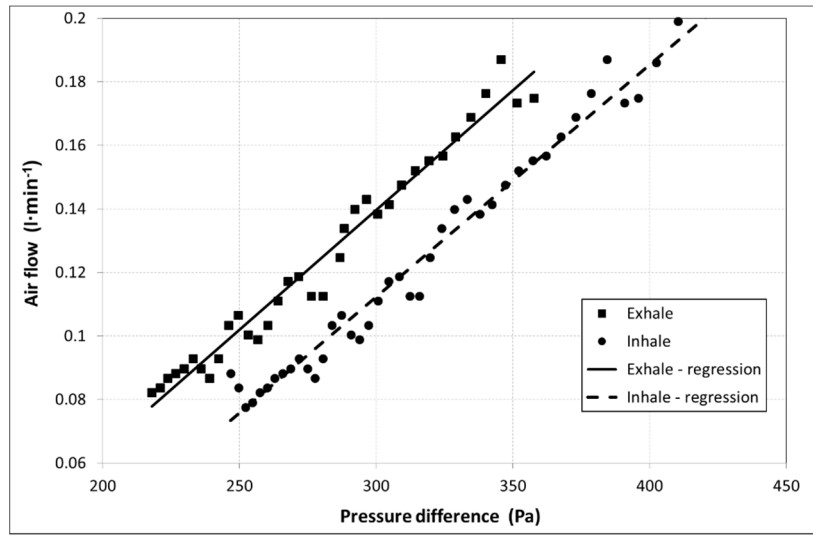

**Figure 11.** A correlation between the flow rate of the air leaking through the skin–facepiece interface and positive pressure (exhalation) inside the mask. A comparison of leakage during inhalation and exhalation.

At a maximum permissible positive pressure (during exhalation) for FMP3 respirators, which represents 300 Pa, the flow rate of the air leaking into the mask was 0.14 L·min$^{-1}$. This value represents 0.09% of the flow rate of 160 L·min$^{-1}$ during inhalation, as defined by the EN 1827 standard.

Considering the maximum pressures for inhalation and exhalation, the measured values of leakage were much lower than the threshold of 2% defined by the standard for protective half masks.

## 5. Conclusions

The application of modern materials with antiviral effects, together with the newly designed shape of a respirator which respects the natural curves of a human face, will facilitate the production of personal protective equipment with a highly efficient filtration system, i.e., with the minimum penetration of aerosol into the air and with a particle size comparable to the size of the coronavirus. The key requirement for PPE certification is to meet several limit values defined in the EN 143, EN 149, and EN 1827 standards, as well as the relevant standards for particulate respirators. The related tests include the determination of the total inward leakage of a half mask, including the inward leakage of the air through the skin–facepiece interface. Based on the assumption that the total penetration of NaCl aerosol resulting from the mask leak is equivalent to the total penetration of the air, a measurement procedure was designed with the use of an indirect method of measurement. The designed measurement system was used to analyse changes in the pressure of the pressure container, and the resulting data was subsequently used to identify the flow rate of the air leaking through the skin–facepiece interface of the mask. The shape of the newly designed half mask, made of Santoprene 8281-45MED, exhibited very good adhesion, while the measured leakage amounted to 0.21%, which represents an air flow rate of 0.2 L·min$^{-1}$ at the maximum permissible negative pressure for FMP3 respirators. The flow rate of the air leaking through the skin–facepiece interface of the half mask during exhalation, at the maximum permissible negative pressure, amounted to 0.14 L·min$^{-1}$. Therefore, considering the maximum pressures for inhalation and exhalation, the measured leakage values were much lower than the threshold of 2% defined in the standard for protective half masks.

**Author Contributions:** Conceptualisation, all authors; methodology, T.B. and M.L.; formal analysis, N.J. and R.D.; visualization, L.T.; validation, J.Ž. and R.H.; writing—review and editing, all authors; supervision, T.B. All authors have read and agreed to the published version of the manuscript.

**Funding:** This work was supported by the Slovak Research and Development Agency under the Contract no. PP-COVID-20-0025. PP-COVID-20-0025-Development and testing of respirators with effective virus degradation by filters containing antiviral materials.

**Institutional Review Board Statement:** Not applicable. Ethical review was not sought for the present study due to the common research ethics standards of the relevant scientific societies as well as survey research. The experiments do not pose any particular risks, i.e., participation in the study does not produce harm or discomfort beyond everyday experience.

**Informed Consent Statement:** Not applicable.

**Data Availability Statement:** Data to support the reported results can be found at the Technical University of Košice, Slovakia; marian.lazar@tuke.sk.

**Conflicts of Interest:** The authors declare no conflict of interest.

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
