# Peer review of "The Determination of the Inward Leakage through the Skin–Facepiece Interface of the Protective Half-Mask"

_applsci, doi:10.3390/app11178042_

Round 1
Reviewer 1 Report
Dear Authors
The paper is very well written and the topic is interesting and important. I will agree to the publication but I would like to ask you to correct the following issues:
- Fig. 10 - please add the y-axis caption on the left side, not on the right side
- The equations no 7 and 8 have the additional number (1), please remove.
- Please add the full title of the project in the "funding" section
Author Response
Dear reviewer,
your comment has been accepted and incorporated into the text.
Answers: Figure 10 was corrected, similar than the unit in the equations 7 and 8. The full title of the project in the funding section was added.
Reviewer 2 Report
Dear authors:
Thank you for this very interesting article. The topic is time-relevant for today’s overall development.
Please, accept my comments as a suggestion for improvement:
It is necessary to check and edit used units in the text according to SI UNITs (formal recommendation).
You can describe in detail the differences between a) and b) illustrations in Figure 3 (Page 6).
Line 90÷ 175 It is suitable to indicate the exact type of valves used in the experiment and the type of control SW.
As well as check the references and the list of references needs to be expanded.
I wish you good luck in your future academic endeavor.
Author Response
Dear reviewer,
your comment has been accepted and incorporated into the text.
Inaccuracies in unit was removed and the units in the text was corrected according to SI UNITs (formal recommendation). For example: line 232-233. The differences between a) and b) illustrations in Figure 3 (Page 6) was added and described – line 202 – 204. Information about the exact type of valves used in the experiment and the type of control SW we indicate in lines 142, 148, 149, 173, 175. References list was expanded.
Reviewer 3 Report
This is a well presented study involving the mechanical and kinetic properties of mask air flow leakage. The manuscript is well written and the experiments well designed. I feel the quality of the graphs must be improved (remove the cross bars, increase axes thickness, use bold legends, use colors if possible) to make the article more attractive and presentable.
It would also be interesting if the authors could include a few lines about the material properties of the polymers which would help in preventing such leakage
Author Response
Dear reviewer,
your comment has been accepted and incorporated into the text.
A formal mistake in Fig. 10 was removed and the image quality has been improved. The material properties of the polymer used in experiment was added, see line 159-168.